# Transformers for Ischemic Stroke Infarct Core Segmentation from Spatio-temporal CT Perfusion Scans

**Lucas de Vries**[1]                                    LUCAS.DEVRIES@AMSTERDAMUMC.NL
**Bart Emmer**[1]                                        B.J.EMMER@AMSTERDAMUMC.NL
**Charles Majoie**[1]                                    C.B.MAJOIE@AMSTERDAMUMC.NL
**Henk Marquering**[1]                              H.A.MARQUERING@AMSTERDAMUMC.NL
**Efstratios Gavves**[2]                                 E.GAVVES@UVA.NL
[1] *Department of Radiology and Nuclear Medicine, Amsterdam UMC, Amsterdam, The Netherlands*
[2] *University of Amsterdam, Amsterdam, The Netherlands*

**Editors:** Under Review for MIDL 2021

## Abstract

The infarct core size is a crucial biomarker for treatment selection for ischemic stroke patients. For this purpose, we present a novel approach to perform infarct core segmentation using CT perfusion (CTP) source data, without ordinary deconvolution analysis. We propose the use of transformers to encode sequential CTP scans in spatial attention maps, which we subsequently use for infarct core segmentation. We report new top results on the ISLES 2018 challenge test data set for infarct core segmentation. This work presents a primary benchmark for infarct core segmentation from CTP source data using transformers.
**Keywords:** CT Perfusion, Transformers, Ischemic stroke, Infarct core segmentation.

## 1. Introduction

Ischemic stroke is one of the deadliest diseases worldwide and a major cause of permanent disability. The infarct core size is a crucial biomarker for treatment selection. In practice, CT perfusion (CTP) imaging is used to determine the infarct core size. Direct interpretation of 4D spatio-temporal CTP source data is challenging due to the low signal-to-noise ratio and, therefore, perfusion parameter maps are typically derived through deconvolution analysis. Deconvolution analysis relies on simplified blood flow models, the choice of an arterial input function, and is sensitive to noise. Consequently, there is a discrepancy in the parameter maps from different CTP analysis software packages (Koopman et al., 2019). The focus of this short paper, therefore, is to train deep learning models for infarct core segmentation directly from CTP source data instead of the handcrafted perfusion maps.

Many studies have shown promising results on infarct core segmentation using spatial perfusion parameter maps as an input to convolutional architectures (Chen et al., 2020). Thus far, studies using only spatio-temporal CTP source data to segment the infarct core are typically less effective (Bertels et al., 2019). A possible reason for this discrepancy could be the sequential nature of CTP source data and the difficulty of accurately encoding spatio-temporal correlations. The sequential data, therefore, triggers us to investigate the use of state-of-the-art sequential models, namely transformers. The primary contributions of this work are: (i) a transformer model to encode spatio-temporal correlations in CTP source data in a valuable spatial attention map, without the selection of an arterial input

function, and (ii) an infarct core segmentation model using the transformer's learned spatial attention map. Our model achieves top performance compared to other methods that only use CTP source data and paves the way for more refined transformer-based architectures.

## 2. Methods

**Data and Preprocessing**   We use data from the ISLES 2018 medical image segmentation challenge. The 94 CTP scans were acquired within 8 hours after stroke onset and the ground truth was delineated on DWI-MRI scans from at most 3 hours after CTP. The scans have a spatial resolution of $256 \times 256$ voxels and consist of 28 to 64 frames. Due to inconsistency in the number of axial slices, we treat each slice (2D + time) as an individual data point for training. We use 4-fold cross-validation, taking into account that all slices from one scan are in the same fold. We use an ensemble of the 4 resulting models to segment the ISLES 2018 test set through voting. To exploit the contra-lateral side of the brain, we flip each scan and register the flipped scan to the original. We retain the flipped scan as an additional channel. We apply temporal smoothing such that every sequence has a length of 32 frames (Bertels et al., 2019), and augment the data with horizontal and vertical flips and rotations.

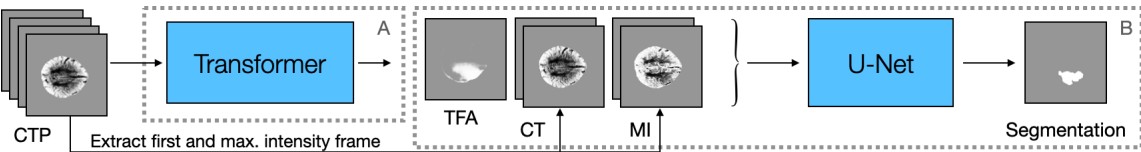

Figure 1: Illustration of our twofold method. Our transformer-generated spatial attention map (TFA) is used together with CTP source data to segment the infarct core.

**Model**   The method in this paper is twofold; we first train a transformer to determine the probability that the 4 central voxels of a patch are infarcted (part A in Figure 1). The transformer's attention map results from applying the model to overlapping patches. The second part (B) of the method segments the infarct core based on the transformer's attention map (TFA), the first frame of the sequence (CT), and the maximum intensity frame (MI).

To train the transformer, we randomly select patches $\mathbf{x} \in \mathbb{R}^{H \times W \times T \times C}$ with $H = W = T = 32$ and $C = 2$, evenly distributed across infarcted and non-infarcted areas. Thereafter, we reshape each input patch to a sequence of flattened patches $\mathbf{x} \in \mathbb{R}^{T \times (HWC)}$ and a learned transformation maps each patch-frame $x_i \in \mathbb{R}^{2048}$ to a latent projection $z_i \in \mathbb{R}^{128}$. Following Vaswani et al. (2017), we learn a 'classification embedding' $z_{cls} \in \mathbb{R}^{128}$, as an extra input to the transformer and we use the output of $z_{cls}$ from the last layer to classify the 4 central pixels. We use additive learned positional embeddings $\mathbf{p} \in \mathbb{R}^{33 \times 128}$. The inputs to our transformer are $\mathbf{z} \in \mathbb{R}^{33 \times 128}$. We use Linformer with 8 heads and 12 layers as our transformer backbone (Wang et al., 2020), and train the model for 150 epochs using the cross-entropy loss and Adam optimizer with a decaying learning rate. The convolutional part of our method is a traditional U-Net architecture with three down-sampling layers. We optimize the network using SGD with momentum and the generalized Dice loss function.

Table 1: Results of our method and models $CE^{0.11}$ and $D^{0.50}$ from Bertels et al. (2019).

|  | $CE^{0.11}$ | $D^{0.50}$ | Threshold | This work | $D^{0.50}$ @ Test | This work @ Test |
|---|---|---|---|---|---|---|
| Dice | **0.50** | 0.46 | 0.43 | 0.49 | 0.38 | **0.42** |
| Precision | 0.43 | **0.51** | 0.39 | 0.48 | **0.47** | 0.44 |
| Recall | 0.58 | 0.40 | 0.62 | **0.66** | 0.44 | **0.53** |

## 3. Results and Discussion

Table 1 presents results from Bertels et al. (2019) and our experiments. For a description of the models $CE^{0.11}$ and $D^{0.50}$, we refer to Bertels et al. (2019). We can threshold the transformer's output to yield a binary segmentation mask. This naive approach results in a fragmented mask due to the locality of our method. Individual small disconnected segments are classified as infarct, leading to high recall and low precision (see Threshold in Table 1). Training a U-Net on the transformer's attention map, the first CTP frame, and maximum intensity frame, solves this and connects and smooths the segmentation mask, leading to an improved average Dice score of 0.49 on the folds. The additional frames convey relevant features for infarct segmentation since our method jointly exploits both sides of the brain. The Dice score drops 8% when we only use the attention map. We achieve a Dice score of 0.42 on the ISLES 2018 test data set, exceeding prior top results by 10% (Bertels et al., 2019).

We do not yet meet the results of convolutional models using spatial maps from deconvolution analysis. Our method is favorable, however, because the transformer yields a deconvolution-free spatial attention map, which is easily interpretable and potentially valuable in a clinical setting. Furthermore, we showed that our simple transformer can encode spatio-temporal correlations. There is still room for improvement in the transformer's design. Therefore, our results provide a primary benchmark and show that transformer-based models are a promising direction for infarct core segmentation from CTP source data.

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
