# OpenReview forum: "Transformers for Ischemic Stroke Infarct Core Segmentation from Spatio-temporal CT Perfusion Scans"
_MIDL.io/2021/Conference/Short — MIDL 2021 Poster_

### Official Review · Reviewer_dy9r · 2021-04-23

**Confidence:** 2
**Final Rating:** 3

**Summary:**

Authors implement transformers to perform infarct core segmentation on CTP images from the ISLES challenge. They have participated in the ISLES challenge, I think as user devrl1 in the live Leaderboard on the test set. Their presented approach appears solid and sound, and shows good performance on the ISLES challenge task.

**Strengths:**

Authors use a dedicated approach (transformers) that takes into account the temporal aspect of the scan. They have validated this on a public dataset with good performance on the leaderboard of the ISLES challenge.

**Weaknesses:**

In my opinion, this short paper format of MIDL is too short for this work. The paper is too compact to be fully understandable, even though the work is very interesting to the MIDL audience.

In the abstract, authors say the "report new top results". This gave me the impression that they achieve first (1st == top) results in the leaderboard on the test data. This does not seem to be the case. Their results are good and impressive, but this should be formulated differently.

I do not fully grasp why authors are only comparing their method to the one of Bertels. Perhaps I missed this rationale / explanation, but there are many methods competing in ISLES. Why single out this one as a comparison?

**Deanonymize Review:**

no

**Detailed Comments:**

Please include a link / username for the ISLES leaderboard.

**Justification Of The Rating:**

The paper appears sound and of interest to the MIDL audience. In my opinion, the short format is too short for this work to be fully understandable. The results are good and can be validated on the external leaderboard.

**Paper Type:**

methodological development

**Special Issue:**

no

---

### Official Review · Reviewer_T9Tw · 2021-04-30

**Confidence:** 3
**Final Rating:** 4

**Summary:**

The paper presents a novel approach to perform infarct core segmentation on CT perfusion source data without deconvolution analysis. The authors evaluate their approach on the ISLES 2018 challenge dataset and show top results (new best method with a Dice Score of 0.42 when not using deconvolutional analysis)

**Strengths:**

- the paper is well written and easy to follow.
- the paper tackles a clinically relevant problem
- even though it is a short paper, detailed information is given on how the method was implemented and trained.
- the method shows promising results (new best method with a Dice Score of 0.42 when not using deconvolutional analysis)

**Weaknesses:**

no example images/results (due to page limit)‏‏‎ ‎‏‏‎ ‎‏‏‎ ‎‏‏‎ ‎‏‏‎ ‎‏‏‎ ‎‏‏‎ ‎‏‏‎ ‎‏‏‎ ‎‏‏‎ ‎‏‏‎ ‎‏‏‎ ‎‏‏‎ ‎‏‏‎ ‎‏‏‎ ‎‏‏‎ ‎‏‏‎ ‎‏‏‎ ‎‏‏‎ ‎‏‏‎ ‎‏‏‎ ‎‏‏‎ ‎‏‏‎ ‎‏‏‎ ‎‏‏‎ ‎‏‏‎ ‎‏‏‎ ‎‏‏‎ ‎‏‏‎ ‎‏‏‎ ‎‏‏‎ ‎‏‏‎ ‎‏‏‎ ‎‏‏‎ ‎‏‏‎ ‎‏‏‎ ‎‏‏‎ ‎‏‏‎ ‎‏‏‎ ‎‏‏‎ ‎‏‏‎ ‎‏‏‎ ‎‏‏‎ ‎‏‏‎ ‎

**Deanonymize Review:**

no

**Justification Of The Rating:**

This paper presents an interesting new approach for a clinically relevant problem and shows promising results (see above). Therefore, I would argue that it should be presented and discussed at MIDL 2021.

**Paper Type:**

both

**Special Issue:**

no

---

### Meta-Review · Program_Chairs · 2021-05-09

**Recommendation:** Accept (Poster)
**Confidence:** 4

**Metareview:**

Reviewers are unanimous in their recommendation to accept this paper.

---

### Decision · Program_Chairs · 2021-05-11

Accept (Poster)